# Indoxyl Sulfate Contributes to mTORC1-Induced Renal Fibrosis via The OAT/NADPH Oxidase/ROS Pathway

**DOI:** 10.3390/toxins13120909

**Published:** 2021-12-18

**Authors:** Takehiro Nakano, Hiroshi Watanabe, Tadashi Imafuku, Kai Tokumaru, Issei Fujita, Nanaka Arimura, Hitoshi Maeda, Motoko Tanaka, Kazutaka Matsushita, Masafumi Fukagawa, Toru Maruyama

**Affiliations:** 1Department of Biopharmaceutics, Graduate School of Pharmaceutical Sciences, Kumamoto University, Kumamoto 8620973, Japan; 214y3001@st.kumamoto-u.ac.jp (T.N.); imafuku@wakayama-med.ac.jp (T.I.); 161p1034@st.kumamoto-u.ac.jp (K.T.); 200y1007@st.kumamoto-u.ac.jp (I.F.); 174p1003@st.kumamoto-u.ac.jp (N.A.); maeda-h@kumamoto-u.ac.jp (H.M.); tomaru@gpo.kumamoto-u.ac.jp (T.M.); 2Department of Nephrology, Akebono Clinic, Kumamoto 8614112, Japan; tanaka@matusita-kai.or.jp (M.T.); kmatu_ipadc@i.softbank.jp (K.M.); 3Division of Nephrology, Endocrinology and Metabolism, Tokai University School of Medicine, Kanagawa 2591193, Japan; fukagawa@tokai-u.jp

**Keywords:** indoxyl sulfate, chronic kidney disease, renal fibrosis, mTORC1, AST-120

## Abstract

Activation of mTORC1 (mechanistic target of rapamycin complex 1) in renal tissue has been reported in chronic kidney disease (CKD)-induced renal fibrosis. However, the molecular mechanisms responsible for activating mTORC1 in CKD pathology are not well understood. The purpose of this study was to identify the uremic toxin involved in mTORC1-induced renal fibrosis. Among the seven protein-bound uremic toxins, only indoxyl sulfate (IS) caused significant activation of mTORC1 in human kidney 2 cells (HK-2 cells). This IS-induced mTORC1 activation was inhibited in the presence of an organic anion transporter inhibitor, a NADPH oxidase inhibitor, and an antioxidant. IS also induced epithelial–mesenchymal transition of tubular epithelial cells (HK-2 cells), differentiation of fibroblasts into myofibroblasts (NRK-49F cells), and inflammatory response of macrophages (THP-1 cells), which are associated with renal fibrosis, and these effects were inhibited in the presence of rapamycin (mTORC1 inhibitor). In in vivo experiments, IS overload was found to activate mTORC1 in the mouse kidney. The administration of AST-120 or rapamycin targeted to IS or mTORC1 ameliorated renal fibrosis in Adenine-induced CKD mice. The findings reported herein indicate that IS activates mTORC1, which then contributes to renal fibrosis. Therapeutic interventions targeting IS and mTORC1 could be effective against renal fibrosis in CKD.

## 1. Introduction

Chronic kidney disease (CKD) is associated with progressive renal fibrosis as renal function declines. A therapeutic strategy targeted to renal fibrosis has become a subject of interest regarding CKD treatment [1]. The activation of mTORC1 (mechanistic target of rapamycin complex 1) in renal tissue was recently reported in renal pathologies such as diabetic kidney disease (DKD) [2], polycystic kidney disease (ADPKD) [3], and focal segmental glomerulosclerosis (FSGS) [4]. In addition, it has been reported that the inhibition of mTORC1 activity suppresses renal fibrosis in various animal models of renal diseases [5,6,7,8,9,10]. Given the above findings, the regulation of renal mTORC1 activity could be a therapeutic target for the treatment of renal fibrosis.

mTORC1 acts as a sensor for intracellular nutrition and stress, and is involved in the maintenance of cellular homeostasis by promoting protein synthesis, mitochondrial synthesis, lipid synthesis, and inhibition of autophagy [11]. It was also demonstrated that the constitutive activation of mTORC1, induced by excess energy, persistent oxidative stress, and fibrosis promoting factor (TGF–β), contribute to the development of renal fibrosis [12,13]. However, the mTORC1 activator and its molecular mechanisms under CKD pathology have still remained unclear.

In CKD, uremic toxins that accumulate in the body due to decreased renal function have been reported to induce various tissue damage [14]. Among these toxins, protein-bound uremic toxins such as indoxyl sulfate (IS), *p*-cresyl sulfate (PCS), phenyl sulfate (PS), hippuric acid (HA), indole acetic acid (IA), kynurenic acid (KA), and 3-carboxy-4-methyl-5-propyl-2-furanpropanoic acid (CMPF) are difficult to remove by hemodialysis because of their strong binding affinity to serum albumin [15]. In fact, recent reports have shown that these uremic toxins are not only involved in the progression of renal damage [16], but are also involved in CKD complications such as cardiovascular disease [17], muscle atrophy and weakness [18], osteoporosis [19], and cognitive decline [20]. The molecular mechanism responsible for the cellular damage caused by these uremic toxins involves the overproduction of reactive oxygen species (ROS) through the activation of NADPH oxidase after they are taken up into various cells via organic anion transporters (OATs) [21,22]. As mentioned above, ROS functions as an activator of mTORC1. Therefore, we hypothesized that the uremic toxin-induced ROS production is responsible for the mTORC1 activation that is observed in the pathogenesis of CKD. However, the relationship among uremic toxins, mTORC1 activation, and renal fibrosis remains unclear.

The objective of this study was to identify the uremic toxins that are involved in mTORC1-induced renal fibrosis. To accomplish this, we first explored the uremic toxins that are involved in mTORC1 activation using human tubular epithelial cells (HK-2 cells), and investigated the molecular mechanisms responsible for this. We then examined the issue of whether the uremic toxin/mTORC1 pathway is involved in epithelial–mesenchymal transition, differentiation into myofibroblasts, and inflammatory responses using tubular epithelial cells, renal fibroblasts, and macrophage cell lines. In in vivo experiments, we investigated the uremic toxin-induced activation of renal mTORC1 using uremic toxin-overloaded mice. Finally, therapeutic interventions targeting the uremic toxin/mTORC1 pathway were performed using adenine-induced CKD mice.

## 2. Results

### 2.1. Effect of Various Uremic Toxins on mTORC1 Activity in Renal Tubular Epithelial Cells (HK-2)

We used human kidney 2 cells (HK-2 cells) to investigate the uremic toxin that promotes mTORC1 activation in renal tubular epithelial cells. We determined the pS6/S6 which is well-known as a surrogate marker of mTORC1 activity by Western blotting analysis [5]. As shown in Figure 1A, mTORC1 activity (pS6/S6) in HK-2 cells was significantly increased when 10% serum from CKD patients was added, as compared to 10% serum from healthy subjects. These findings suggested that uremic toxins present in the serum from CKD patients could be involved in mTORC1 activation.

We next evaluated the effects of seven representative uremic toxins (IS, PCS, PS, HA, IA, KA, and CMPF) on mTORC1 activation in HK-2 cells. As shown in Figure 1B, among the seven uremic toxins that were examined, only IS was found to activate mTORC1 (pS6/S6). We also found that the IS activation of mTORC1 was concentration-dependent, starting at around 0.1 mM, a concentration that was observed in CKD patients (Figure 1C). In fact, we measured IS concentrations in the serum from healthy subjects as well as the CKD patients described in Figure 1A, and evaluated the relationship between serum IS concentration and mTORC1 activity. As shown in Figure 1D, there was a significant positive correlation between the serum concentration of IS and mTORC1 activity. These results indicate that, among the various uremic toxins, IS appears to be the major activator of mTORC1 in tubular epithelial cells.

### 2.2. Molecular Mechanism of IS-Induced mTORC1 Activation

We investigated the molecular mechanism of mTORC1 activation induced by IS. As shown in Figure 1E, IS-induced mTORC1 activation in HK-2 cells was markedly inhibited in the presence of OATs inhibitor (probenecid), NADPH oxidase inhibitor (DPI), and antioxidant (AsA). These results suggest that IS is taken up into cells via OATs, where it then activates mTORC1 by producing ROS associated with the activation of NADPH oxidase.

### 2.3. Involvement of IS-Induced mTORC1 Activation in Fibrotic Responses of Tubular Epithelial Cells, Renal Fibroblasts, and Macrophages

It has been reported that the epithelial–mesenchymal transition (EMT) of tubular epithelial cells, the differentiation of fibroblasts to myofibroblasts, and the inflammatory response of macrophages are involved in the development of renal fibrosis [23]. As shown in Figure 2A, in HK-2 cells (human tubular epithelial cells), IS induced the activation of mTORC1, caused a decrease in the expression of E-cadherin (epithelial marker), and increased the expression of α-SMA (mesenchymal marker), and these effects were suppressed by the presence of an mTORC1 inhibitor (rapamycin). In NRK-49F (rat renal fibroblast), IS also induced the activation of mTORC1 and increased the expression of α-SMA and COL1A1 (myofibroblast markers), and these effects were suppressed in the presence of rapamycin (Figure 2B). In Figure 2C, in differentiated THP-1 (dTHP1: human monocyte-derived macrophages) cells, IS induced the activation of mTORC1 and increased the expression of pro-inflammatory cytokines (TNF-α and IL-6), and these effects were also suppressed by the presence of rapamycin. These results suggest that IS induces the epithelial–mesenchymal transition of tubular epithelial cells, myofibroblast differentiation from fibroblasts, and inflammatory responses in macrophages through mTORC1 activation.

### 2.4. Effect of Exogenous IS-Overload on mTORC1 Activity in Renal Tissue of Healthy Mice

To confirm that IS contributes to renal mTORC1 activity under in vivo conditions, IS (100 mg/kg/day) was intraperitoneally administered to healthy mice for 3 days (Figure 3A). Increases in both the plasma and renal levels of IS were observed after IS-overload (Figure 3B). As shown in Figure 3C, IS-overload significantly increased renal mTORC1 activity. These data suggest that increased renal IS could contribute to renal mTORC1 activation, even under in vivo conditions.

### 2.5. Therapeutic Intervention Involving Targeting the IS/mTORC1 Pathway in CKD Mice

We next examined the issue of whether therapeutic interventions targeting IS and the mTORC1 pathways serve to inhibit CKD-associated renal fibrosis. AST-120 suppresses the absorption of indole, an IS precursor, from the intestinal tract, thereby lowering the IS levels in the body. As shown in Figure 4A, CKD was induced by feeding 0.2% adenine for 2 weeks, followed by therapeutic intervention with an oral adsorbed charcoal (AST-120) or an mTORC1 inhibitor (rapamycin) for 3 weeks. As shown in Figure 4B, the increased IS levels in plasma and renal tissues of the CKD mice was significantly suppressed by the AST-120 treatment, but not by the rapamycin treatment. In addition, the activation of mTORC1 in the renal tissue of CKD mice was significantly suppressed by AST-120 or the rapamycin treatment (Figure 4C). In the renal tissues of CKD mice, the down-regulation of E-cadherin, up-regulation of α-SMA and COL1A1 (Figure 4D), and the up-regulation in the mRNA expression of inflammatory cytokines (TNF-α and IL-6) (Figure 4E) were observed, and these changes were significantly suppressed by the administration of AST-120 or rapamycin. The fibrotic areas (staining with Picrosirius red) and increased hydroxyproline levels in the renal tissue in CKD mice were also significantly suppressed by the administration of AST-120 or rapamycin (Figure 4F). These results indicate that IS contributes to renal fibrosis by activating mTORC1 in renal tissue under CKD conditions.

## 3. Discussion

Although strategies targeting mTORC1 signaling in CKD treatment are expected, there are still many unclear points concerning the mTORC1 activator and its molecular mechanism in cases of CKD pathology. In this study, we found that IS, a protein-bound uremic toxin, activates mTORC1 via the OAT/NADPH oxidase/ROS pathway in tubular epithelial cells, suggesting that the IS/mTORC1 pathway might well be involved in epithelial–mesenchymal transition of tubular epithelial cells, the differentiation of fibroblasts into myofibroblasts and the inflammatory response of macrophages. Therapeutic interventions targeting IS and mTORC1 in a mouse model of CKD were found to suppress renal fibrosis, suggesting that therapeutic strategies targeting the IS/mTORC1 pathway for the treatment of renal fibrosis, as observed in CKD, might be useful.

mTORC1 enhances the protein synthesis system after recognizing branched-chain amino acids (BCAAs) and insulin/insulin-like growth factor 1 (IGF-1) signaling. In CKD, excessive renal mTORC1 activation induced by hypoxia, oxidative stress, and fibrosis promoting factor (TGF-β), in addition to the accumulation of glucose or palmitate in renal tissues, has been reported [12,24,25,26,27]. This higher renal mTORC1 activation was reported to be involved in CKD progression [2,3,4]. However, the issue of whether protein-bound uremic toxins, which are associated with renal and life prognosis in CKD patients [28], cause mTORC1 activation has not been examined in detail. The findings reported in this study revealed that, among the seven protein-binding uremic toxins, only IS activated mTORC1 using HK-2 cells (Figure 1B). In addition, a significant positive correlation was observed between mTORC1 activity and IS levels in the serum from CKD patients (Figure 1D), indicating that, IS is a factor in the activation of mTORC1. Interestingly, when IS was added to cultures of HK-2 cells, mTORC1 was significantly activated when the concentration of IS was 100 μM or higher (Figure 1C). Furthermore, the concentration of IS in the serum from the CKD patients ranged from 100 to 300 μM (Figure 1D). In fact, serum IS levels in uremic patients have been reported to be increased to levels as high as 1 mM [14]. These findings suggest that mTORC1 activation in the tubular epithelium could readily occur in CKD patients with higher IS levels in the body.

Among the seven protein-bound uremic toxins, we found that only IS activated mTORC1. Regarding the biological activity of IS, it was reported that IS acts as a ligand for the aromatic hydrocarbon receptor (AhR) [29] and the epidermal growth factor receptor (EGFR) [30] to produce ROS. IS also activates the renin-angiotensin-aldosterone system (RAAS), with the production of ROS [31]. In addition, the activation of NADPH oxidase was found to be involved in these ROS production pathways [29,32,33]. In fact, our data showed that the IS-induced mTORC1 activation was significantly suppressed in the presence of an NADPH oxidase inhibitor (DPI) (Figure 1E). Furthermore, among the seven protein-bound uremic toxins, only IS promoted the production of significant levels of ROS in HK-2 cells (Appendix A). Therefore, the possibility that AhR, EGFR, and RAAS are involved in the function of NADPH oxidase and mTORC1 activation cannot be excluded. However, in our experiments using HK-2 cells, the involvement of AhR appeared to be minor, because IS-induced ROS production and mTORC1 activation were not inhibited in the presence of an AhR inhibitor (data not shown). Therefore, further investigations will be needed to clarify the contribution of EGFR and the RAAS pathway in the IS-induced activation of mTORC1.

The epithelial–mesenchymal transition (EMT) in tubular epithelial cells, the differentiation of renal fibroblasts into myofibroblasts, and inflammatory response of macrophages are all involved in the development of renal fibrosis [23]. Kim et al. previously reported that IS promotes hypertrophy, apoptosis, TGF-β production, EMT in tubular cells [34,35]. Milanesi et al. reported that IS induced the production of the monocyte chemotactic factor (MCP-1) and TGF-β production in fibroblasts [36], and Nakano et al. also demonstrated that IS contributed to the production of pro-inflammatory cytokines in macrophages and their differentiation into pro-inflammatory macrophages (M1) [17,37]. Our findings indicate that IS acts in a pro-fibrotic manner via the activation of mTORC1 in tubular epithelial cells, renal fibroblasts, and macrophages (Figure 2). These in vitro data were confirmed by the in vivo data showing that AST-120 and the administration of rapamycin suppressed the decreased expression of E-cadherin, elevated levels of α-SMA and COL1A1, and the increased expression of inflammatory cytokines (TNF-α and IL-6) in renal tissues, as observed in CKD model mice without affecting renal function. These collective in vitro and in vivo findings suggest that IS accumulation in renal tissues contributes to renal fibrosis via the activation of mTORC1 in tubular epithelial cells, renal fibroblasts, and macrophages.

In this study, the CKD model mice were fed 0.2% adenine for a period of 2 weeks and were then switched to a normal diet to reflect the early stage of CKD (Appendix A). Therefore, in a future study, evaluating the effect of AST-120 and rapamycin in a more severe CKD model will be needed. We focused on IS as a uremic toxin that the serum level is decreased by AST-120 administration. However, further studies may be needed to evaluate the other mechanisms that AST-120 affected. Milanesi et al. have shown that IS contributes to renal fibrosis by inducing heat shock protein 90 (HSP90) in renal fibroblasts [36]. Therefore, HSP90 pathway may also be involved in IS-induced mTORC1 activation. In addition, IS also contributes to the anemia observed in CKD [38]. In our study, the administration of AST-120 recovered the decrease in red blood cell count and hemoglobin level as observed in CKD mice (Appendix A). These findings suggest that anemia may associated with IS-iduced mTORC1 activation in CKD.

We found that accumulated IS promotes renal fibrosis via mTORC1 under CKD conditions. These results suggest that targeting IS/mTORC1 could be a useful therapeutic strategy for the treatment of CKD. In fact, the use of a low-protein diet [39] and AST-120 to reduce the accumulation of IS and other uremic toxins has been reported to delay the progression of renal function decline in CKD patients [40]. In addition, it has been reported that SGLT2 inhibitors, that have been recently approved for the treatment of CKD, have a nephroprotective effect by inhibiting mTORC1 by increasing ketone levels in renal tissue [5]. These findings suggest that the combined use of a low protein diet, AST-120 and SGLT2 inhibitors in CKD patients with high serum IS levels may delay the progression of CKD. Although mTORC1 inhibitors (rapamycin and everolimus, etc.) are attracting attention as a treatment for renal diseases, there have been some positive or negative clinical outcomes [3,41,42,43]. Further clinical study should be needed in the future. From our data, serum IS levels may be a surrogate marker for mTORC1 activity in CKD patients.

## 4. Conclusions

Among several uremic toxins, IS was found to be the major contributor to the activation of renal mTORC1. Our findings support the possibility of a therapeutic intervention targeting IS/mTORC1 signaling against renal fibrosis.

## 5. Materials and Methods

### 5.1. Chemicals and Materials

Indoxyl sulfate (IS), indole acetic acid (IA), kynurenic acid (KA), 3-carboxy-4-methyl-5-propyl-2-furanpropanoic acid (CMPF), probenecid (Prob) and diphenylene iodonium (DPI) were purchased from Sigma-Aldrich (St Louis, MO, USA). Hippuric acid (HA) and ascorbic acid (AsA) were purchased from Nacalai Tesque (Kyoto, Japan). Phenyl Sulfate (PS) was purchased from Tokyo Chemical Industry (Tokyo, Japan). Rapamycin was purchased from LC laboratories (Woburn, MA, USA). *p*-Cresyl sulfate (PCS) was synthesized according to a method reported by Feigenbaum et al. [44], and its purity was confirmed by nuclear magnetic resonance spectroscopy [45]. AST-120 was provided by Kureha Corporation (Tokyo, Japan). All methods were performed in accordance with the relevant guidelines and regulations, and were approved by Kumamoto University.

### 5.2. Cell Culture

Human renal proximal tubule epithelial cells (HK-2 cells) and rat renal interstitial fibroblast (NRK-49F cells) were cultured in DMEM/Ham’s F-12 medium (FUJIFILM Wako Pure Chemical Corporation, Osaka, Japan) with 10% FBS (Capricorn scientific, Ebsdorfergrund, Germany) and 1% antibiotic/antifungal mixture solution (100 U/mL penicillin, 100 μg/mL streptomycin, 0.25 μg/mL amphotericin B; Nacalai Tesque, Kyoto, Japan) at 37 °C in an atmosphere containing 5% CO_2_. Human monocytes (THP-1 cells) were cultured in RPMI 1640 (Gibco, Grand Island, NY, USA) with 10% FBS and 1% antibiotic/antifungal mixture solution. THP-1 cells were seeded and differentiated for 48 h by adding 50 nM phorbol 12-myristate 13-acetate (Sigma-Aldrich, St Louis, MO, USA). These cells were seeded in 6-well plates in appropriate cell numbers and used for the experiments (see legends for Figures).

### 5.3. Cell Experiments with Serum Samples from CKD Patients

Sixteen CKD patients between the ages of 69 and 86 years (mean 81.0 ± 1.6 years) who were undergoing dialysis and who had been admitted to the Department of Nephrology at the Akebono Clinic in Japan in October 2018 were enrolled in this study. As controls, healthy volunteers (4 males and 2 females) aged 21–29 years (mean 23.6 ± 0.7 years) were enrolled. The experimental method was described previously [42]. This study was conducted in accordance with the Declaration of Helsinki and was approved by the Ethics Committee of the Faculty of Life Sciences, Kumamoto University (approval number 1578).

### 5.4. Western Blot Analysis

Western blot analysis was performed as previously described [46]. Blots were transferred to PVDF membrane (Immobilon-P; Millipore, Billerica, MA, USA) and then incubated with primary antibodies at 4 °C for 24 h. The membrane was then incubated with horseradish peroxidase-conjugated secondary antibody for 1 hour at room temperature. The intensity of each band was detected using a Vilber-Lourmat FUSION instrument (M&S Instruments, Osaka, Japan) and quantified using ImageJ software. Densitometry intensity was normalized by the expression levels of β-actin. The primary and secondary antibodies used in the Western blot analyses were described in Appendix A.

### 5.5. Quantitative RT-PCR

Quantitative RT-PCR was performed [47]. Primers used for mRNA detection are listed in Appendix A. The threshold cycle (Ct) value for each gene was normalized by subtracting the Ct value calculated for GAPDH. The mRNA expression levels were calculated using the 2^−ΔΔCt^ method.

### 5.6. HPLC Analysis

The concentration of IS in plasma and renal tissues were determined using a previously described HPLC method [48]. HPLC analyses were performed on a Shiseido (Tokyo, Japan) CAPCELL PAK C18 column (150 × 2.0 mm, 5 μm). IS was detected by a fluorescence monitor with excitation and emission wavelengths set at 280 nm and 375 nm, respectively.

### 5.7. Animal Experiments

The study was carried out in compliance with the ARRIVE guidelines. The Animal Care and Use Committee of Kumamoto University approved the protocols for all animal experiments (A2021-021). ICR mice (male, 5 weeks, Japan SLC, Inc., Shizuoka, Japan) were maintained in a room under controlled temperature conditions with a 12 h light and 12 h dark cycle (light 8 am–8 pm) and were freely provided with food and water. CKD model mice were produced by feeding 0.2 *w*/*w*% adenine (FUJIFILM Wako Pure Chemical, Osaka, Japan) mixed with powder diet (CE-2: CLEA, Tokyo, Japan) to mice for 2 weeks. After producing the CKD model mice, they were randomized by blood urea nitrogen (BUN) and body weight and were assigned to AST-120 (charcoal oral absorbent, 8 *w*/*w*% in powder diet) or rapamycin (100 mg/kg/day, dissolved in DMSO, daily administered intraperitoneally) treatment groups. Control mice and untreated CKD mice received a normal diet. For IS-overloaded mice, the mice were intraperitoneally administered with an IS dissolved in saline at 100 mg/kg for 3 days. As a control, saline was intraperitoneally administered to the control mice for 3 consecutive days.

### 5.8. Histological Analysis

The kidney tissue was fixed in 10% neutral formalin buffer for 48 h and then embedded in paraffin. The resulting kidney blocks were cut into 2 μm sections and analyzed for fibrosis by staining with picrosirius red staining. Images and quantitative analysis were performed using a Keyence BZ-X710 microscope (Keyence, Osaka, Japan). The quantification of picrosirius red positive areas was calculated as previously described [49]. The quantification involved calculations from 10 randomly selected fields from each sample.

### 5.9. Hydroxyproline Assay

Renal hydroxyproline was measured using the method described previously [50].

### 5.10. Statistical Analysis

Correlation data were evaluated using Pearson correlation analysis. The means of the data for the two groups were compared using the Student’s *t*-test. The means of more than two groups were compared using analysis of variance followed by Tukey’s multiple comparisons. All data are presented as the mean ± SE. *p* < 0.05 was considered to be a statistically significant difference.

## Figures and Tables

**Figure 1 toxins-13-00909-f001:**
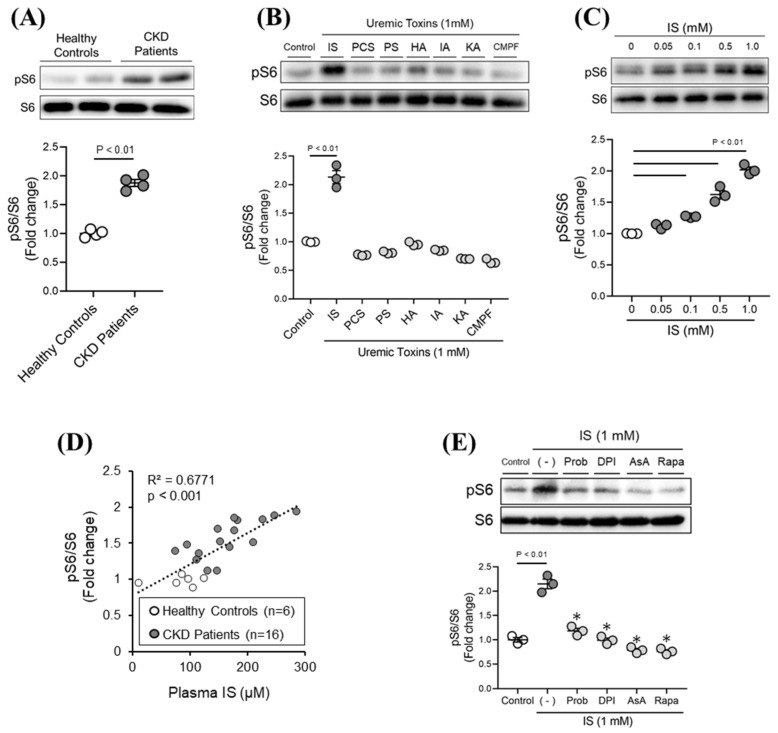
IS activates mTORC1 via the OAT/NADPH oxidase/ROS pathway. (**A**) mTORC1 activity (pS6/S6) in HK-2 cells was measured after incubation with 10% serum obtained from healthy subjects or CKD patients for 12 h. (**B**) The effect of seven uremic toxins (IS, PCS, PS, HA, IA, KA, and CMPF: 1 mM) on mTORC1 activity in HK-2 cells for 12 h. (**C**) Dose-dependent effect of IS on mTORC1 activity in HK-2 cells for 12 h. (**D**) Correlation between IS concentration in plasma from healthy controls (*n* = 6) or CKD patients (*n* = 16) and mTORC1 activity in HK-2 cells. (**E**) Effect of an organic anion transporter (OAT) inhibitor (probenecid: Prob (0.5 mM)), NADPH oxidase inhibitor (diphenylene iodonium: DPI (5 μM)), antioxidant (ascorbic acid: AsA (0.5 mM)) or mTORC1 inhibitor (rapamycin: Rapa (5 nM)) on IS (1 mM)-induced mTORC1 activity in HK-2 cells. The cells were exposed to each inhibitor (Prob, DPI, AsA, and Rapa) for 1 hour and then incubated with IS for 12 h. * *p* < 0.05 compared with IS in the absence of inhibitor. Data are expressed as the means ± SEM (*n* = 3–4).

**Figure 2 toxins-13-00909-f002:**
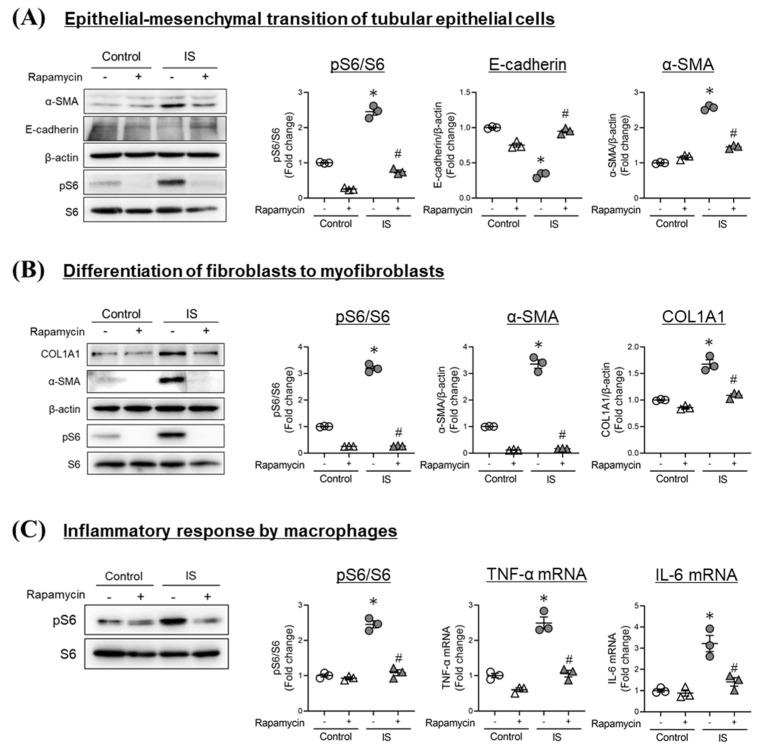
IS induces fibrotic responses of tubular epithelial cells, renal fibroblasts, and macrophages via mTORC1 signaling. (**A**) HK-2 cells were exposed to rapamycin (5 nM) for 1 hour and then incubated with IS (1 mM) for 12 h. p-S6, S6, E-cadherin, α-SMA, and β-actin expression were determined by Western blot analysis. (**B**) NRK-49F cells were exposed to rapamycin (5 nM) for 1 h and then incubated with IS (1 mM) for 12 h. p-S6, S6, α-SMA, COL1A1, and β-actin expression were determined by Western blot analysis. (**C**) Differentiated THP-1 cells were exposed to rapamycin (5 nM) for 1 hour and then incubated with IS (1 mM) for 1 hour. p-S6 and S6 expression were determined by Western blot analysis. TNF-α, IL-6 mRNA expressions were determined by quantitative RT-PCR. Data are expressed as the means ± SEM (*n* = 3). * *p* < 0.05 compared with control in the absence of rapamycin; # *p* < 0.05 compared with IS in the absence of rapamycin.

**Figure 3 toxins-13-00909-f003:**
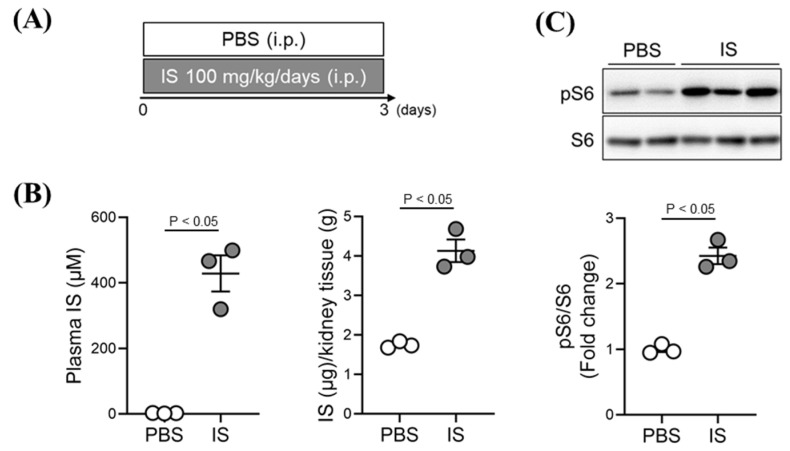
Effect of exogenous IS-overload on mTORC1 activity in renal tissue of healthy mice. Healthy mice were administrated with IS (100 mg/kg/day, intraperitoneal administration) for 3 consecutive days. The control mice were administrated with the same volume of PBS. At three hours after the final administration, the mice were anesthetized, then blood and kidneys were collected. IS levels in (**A**) plasma and (**B**) kidney were measured by HPLC methods. (**C**) mTORC1 activity (pS6/S6) in the kidney at three hours after the final IS administration, was determined by Western blotting. Data are expressed as the means ± SEM (*n* = 3).

**Figure 4 toxins-13-00909-f004:**
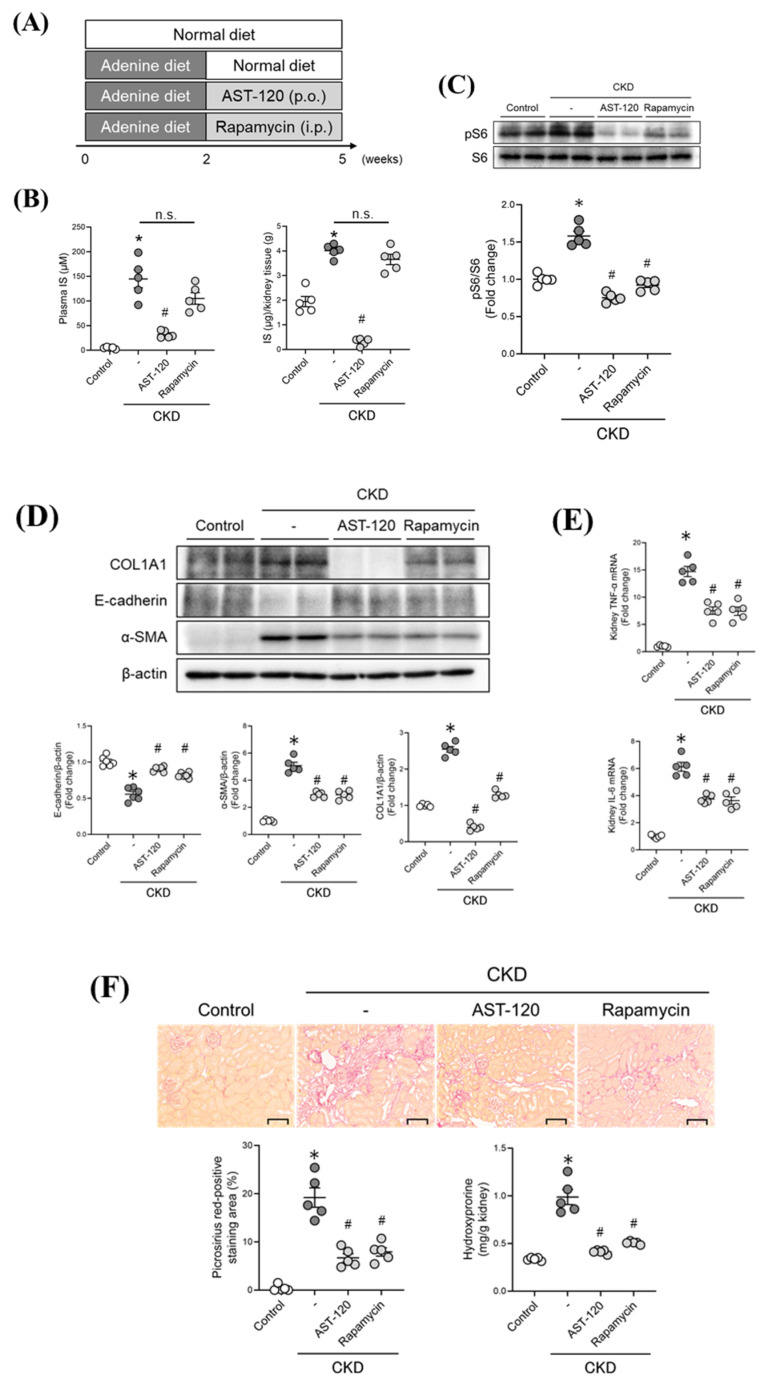
Therapeutic intervention targeting the IS/mTORC1 pathway in CKD mice. (**A**) Experimental protocol for the effect of AST-120 or rapamycin on CKD mice. After randomization at 2 weeks after feeding a 0.2% adenine-containing diet, the AST-120 group was fed an 8% AST-120 containing diet and the rapamycin group received rapamycin (1 mg/kg) intraperitoneally daily. (**B**) Effect of AST-120 or rapamycin on IS levels in the plasma and kidney of CKD mice. (**C**) mTORC1 activity (pS6/S6) in the kidney of CKD mice. (**D**) E-cadherin, α-SMA, COL1A1, and β-actin protein expressions in the kidney of CKD mice. (**E**) TNF-α, IL-6 mRNA expression in the kidney of CKD mice. (**F**) Representative photomicrographs and quantification of Picrosirius red-stained kidney sections of CKD mice are shown. Original magnifications: ×200. Scale bars represent 100 μm. Hydroxyproline content in the kidney of CKD mice was measured. Data are expressed as the means ± SEM (*n* = 5), * *p* < 0.05 compared with control in the absence of rapamycin; # *p* < 0.05 compared with IS in the absence of rapamycin.

## Data Availability

Data are contained within the article or Appendix A. The data presented in this study are available in this article.

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
