# Peer review of "Indoxyl Sulfate Contributes to mTORC1-Induced Renal Fibrosis via The OAT/NADPH Oxidase/ROS Pathway"

_toxins, 2021, doi:10.3390/toxins13120909_

Round 1

Reviewer 1 Report

I was very pleased reading your research paper. It is very well written and opens door to very intriguing possibilities of targeting mTOR pathway in delaying CKD progression or complications of ESRD. Few comments and suggestions as below:

Line 8 - abbreviation and complete form are both written as HK-2 cells - should be human kidney 2 cells (HK-2 cells), instead 

Line 29 - would remove "etc" at the end of the sentence as adds no more to the claim 

Line 40 - should be "have still remained unclear" instead of "remains"

Line 70 - even though it is known to be a good surrogate marker of mTOR activity, needs further explanation or reference as to why pS6/S6 was used

Line 73 - would recommend to use other word than "including", such as "Present"

Line 101 - would remove "s" from OATs, and remove "an" before NADHP

Lines 149-165 - needs further explanation as to why AST-120 was used and how it had the effects demonstrated in the experiment. Is this through by merely reducing the levels of IS and the downstream effects are a reflection of reduced levels, or there are other possible mechanisms too. Also, would be interesting to know how does AST-120 reduce IS level even after days of IS overloading, and its concentration within the kidney tissue. 

In figures please use consistency is reported data expression. Would recommend to adjust in every figure as "data are expressed as the mean/means ... "

Lines 180-181 - would be better to word the sentence otherwise - for example, "although strategies targeting mTORC1 signaling in CKD treatment are expected"

Line 186 - would change to "inflammatory response of macrophages"

Line 191 - IGF-1 was first introduced, and before abbreviation, would need definition of its full name 

Line 193 - would just say accumulation or glucose or palmitate instead of continuing with palmitate accumulation in renal tissues, as sounds redundant 

Line 197 - "remains" is unnecessary and likely a mistake when making changes in the text

Line 198 - would instead say "among the seven protein-bound..."

And one last question, there is much data available on IS and its effects. You have not mentioned HSP-90 pathway in your research. Is this because you have not explored this mechanism, or it was not considered relevant in your experiment and obtained data? 

Also, might be prudent to do more clinical literature review in discussion when you open up the possibilities for future treatment, that there have been some successes and some negative trials (e.g. rapamycin and ADPKD), just to be more thorough. 

Author Response

Response to Reviewers' Comments

We examined the comments made by the referees carefully and prepared a revised version of our paper, in which the comments made by the referees have been addressed. I hope that the present, improved version of our manuscript will now be found to be acceptable for publication in Toxins.

Reviewer 1

Comment-1

Line 8 - abbreviation and complete form are both written as HK-2 cells - should be human kidney 2 cells (HK-2 cells), instead.

Reply-1

As the reviewer suggested, the phrase “HK-2 cells” was changed to “” in the Abstract and Results section of the revised manuscript. (page 1, line 21) and (page 2, line 53)

Comment-2

Line 29 - would remove "etc" at the end of the sentence as adds no more to the claim

Reply-2

As the reviewer suggested, “etc” was removed from the Introduction section of the revised manuscript. (page 1, line 41)

Comment-3

Line 40 - should be "have still remained unclear" instead of "remains"

Reply-3

As the reviewer suggested, “remains” was changed to “remained” in the Introduction section of the revised manuscript. (page 2, line 54)

Comment-4

Line 70 - even though it is known to be a good surrogate marker of mTOR activity, needs further explanation or reference as to why pS6/S6 was used.

Reply-4

As the reviewer suggested, the following sentences and reference were added in the Results section of the revised manuscript. “We determined the pS6/S6 which is well-known as a surrogate marker of mTORC1 activity by western blotting analysis.” (page 2, line 84-line 86)

Reference

  1. Tomita I, Kume S, Sugahara S, et al. SGLT2 Inhibition Mediates Protection from Diabetic Kidney Disease by Promoting Ketone Body-Induced mTORC1 Inhibition. Cell Metab 2020;32(3):404-419.e406

Comment-5

Line 73 - would recommend to use other word than "including", such as "Present"

Reply-5

As the reviewer suggested, “including” was changed to “present” in the Results section of the revised manuscript. (page 2, line 88)

Comment-6

Line 101 - would remove "s" from OATs, and remove "an" before NADHP

Reply-6

As the reviewer suggested, the “s” was removed from OATs and the “an” before OAT, NADPH oxidase or antioxidant was removed in the Results section of the revised manuscript. (page 3, line 121-line 122)

Comment-7

Lines 149-165 - needs further explanation as to why AST-120 was used and how it had the effects demonstrated in the experiment. Is this through by merely reducing the levels of IS and the downstream effects are a reflection of reduced levels, or there are other possible mechanisms too. Also, would be interesting to know how does AST-120 reduce IS level even after days of IS overloading, and its concentration within the kidney tissue.

In figures please use consistency is reported data expression. Would recommend to adjust in every figure as "data are expressed as the mean/means ... "

Reply-7

As the reviewer suggested, the following sentence was added in the Results section of the revised manuscript. “AST-120 suppresses the absorption of indole, an IS precursor, from the intestinal tract, thereby lowering the IS levels in the body.” (page 5, line 177-line 179)

As the reviewer suggested, the following sentences were also added in the Discussion section of the revised manuscript. “We focused on IS as a uremic toxin that the serum level is decreased by AST-120 administration. But, further studies may be needed to evaluate the other mechanisms that AST-120 affected.” (page 8, line 287-line 289)

As mentioned above, AST-120 suppresses the absorption of indole, an IS precursor, from the intestinal tract, thereby lowering the IS levels in the body. Therefore, AST-120 could not decrease the serum IS level after intraperitoneally IS administration.

Finally, as the reviewer suggested, the following sentence was added in the Figure Legends section of the revised manuscript. “Data are expressed as the means ± SEM.” (page 3, line 116-line 117), (page 5, line 157), (page 5, line 173-line 174), and (page 7, line 210-line 211)

Comment-8

Lines 180-181 - would be better to word the sentence otherwise - for example, "although strategies targeting mTORC1 signaling in CKD treatment are expected"

Reply-8

As the reviewer suggested, the phrase “Although the strategies targeting mTORC1 signaling in CKD treatment have been expected” was changed to “Although strategies targeting mTORC1 signaling in CKD treatment are expected” in the Discussion section of the revised manuscript. (page 7, line 213)

Comment-9

Line 186 - would change to "inflammatory response of macrophages"

Reply-9

As the reviewer suggested, the phrase “inflammation of macrophages” was changed to “inflammatory response of macrophages” in the Discussion section of the revised manuscript. (page 7, line 219)

Comment-10

Line 191 - IGF-1 was first introduced, and before abbreviation, would need definition of its full name

Reply-10

As the reviewer suggested, the word “IGF-1” was changed to “insulin-like growth factor 1 (IGF-1)” in the Discussion section of the revised manuscript. (page 7, line 224)

Comment-11

Line 193 - would just say accumulation or glucose or palmitate instead of continuing with palmitate accumulation in renal tissues, as sounds redundant

Reply-11

As the reviewer suggested, the phrase “the accumulation of glucose or palmitate accumulation in renal tissues” was changed to “the accumulation of glucose or palmitate in renal tissues” in the Discussion section of the revised manuscript. (page 7, line 226)

Comment-12

Line 197 - "remains" is unnecessary and likely a mistake when making changes in the text

Reply-12

As the reviewer suggested, the word “remains” was removed in the Discussion section of the revised manuscript. (page 7, line 230)

Comment-13

Line 198 - would instead say "among the seven protein-bound..."

Reply-13

As the reviewer suggested, the phrase “among seven protein-binding uremic toxins” was changed to “among the seven protein-binding uremic toxins” in the Discussion section of the revised manuscript. (page 7, line 231)

Comment-14

And one last question, there is much data available on IS and its effects. You have not mentioned HSP-90 pathway in your research. Is this because you have not explored this mechanism, or it was not considered relevant in your experiment and obtained data?

Also, might be prudent to do more clinical literature review in discussion when you open up the possibilities for future treatment, that there have been some successes and some negative trials (e.g. rapamycin and ADPKD), just to be more thorough.

Reply-14

We have not explored the HSP-90 pathway. But as the reviewer pointed out, HSP-90 pathway may be included in IS-induced mTORC1 activation (Milanesi S et al. Oxid Med Cell Longev. 2019).

Therefore, the following sentences and reference were added in the Discussion section of the revised manuscript. (page 8, line 289-line 290)

“Milanesi et al. have shown that IS contributes to renal fibrosis by inducing heat shock protein 90 (HSP90) in renal fibroblasts [36]. Therefore, HSP90 pathway may also be involved in IS-induced mTORC1 activation” (Milanesi S et al. Oxid Med Cell Longev. 2019).

Reference

  1. Milanesi S, Garibaldi S, Saio M, et al. Indoxyl Sulfate Induces Renal Fibroblast Activation through a Targetable Heat Shock Protein 90-Dependent Pathway. Oxid Med Cell Longev 2019;2019:2050183

In addition, as the reviewer pointed out, there have been some successes and some negative trials (e.g. rapamycin or everolimus for ADPKD).

Therefore, the following sentences and reference were added in the in the Discussion section of the revised manuscript. (page 8, line 305-line 308)

“Although mTORC1 inhibitors (rapamycin and everolimus etc.) are attracting attention as a treatment for renal diseases, there have been some positive or negative clinical outcomes (Shillingford JM et al. Proc Natl Acad Sci U S A. 2006; Braun WE et al. Clin J Am Soc Nephrol. 2014; Walz G et al. N Engl J Med. 2010; Serra AL et al. N Engl J Med. 2010). Further clinical study should be needed in the future. From our data, serum IS levels may be a surrogate marker for mTORC1 activity in CKD patients.”

References

  1. Shillingford JM, Murcia NS, Larson CH, et al. The mTOR pathway is regulated by polycystin-1, and its inhibition reverses renal cystogenesis in polycystic kidney disease. Proc Natl Acad Sci U S A 2006;103(14):5466-5471
  2. Braun WE, Schold JD, Stephany BR, et al. Low-dose rapamycin (sirolimus) effects in autosomal dominant polycystic kidney disease: an open-label randomized controlled pilot study. Clin J Am Soc Nephrol 2014;9(5):881-888
  3. Walz G, Budde K, Mannaa M, et al. Everolimus in patients with autosomal dominant polycystic kidney disease. N Engl J Med 2010;363(9):830-840
  4. Serra AL, Poster D, Kistler AD, et al. Sirolimus and kidney growth in autosomal dominant polycystic kidney disease. N Engl J Med 2010;363(9):820-829

Reviewer 2 Report

The authors submit an original research article entitled "Indoxyl sulfate contributes to mTORC1-induced renal fibrosis via the OAT/NADPH oxidase/ROS pathway ". Indoxyl sulfate (IS) causes significant activation of mTORC1 in HK-2 cells, inhibited in the presence of an organic anion transporter inhibitor, a NADPH oxidase inhibitor, and an antioxidant. IS also induces epithelial-mesenchymal transition of tubular epithelial cells.  in
vivo , IS-overload activates mTORC1 in the mouse kidney. The administration of AST-120 or rapamycin targeted to IS or mTORC1 ameliorated renal fibrosis in Adenine-induced CKD mice.

Major

please give specific number for animal experiment authorization. What was the sex of animals used? please justify the choice;

Please give more information on collection and use of serum samples from CKD and normal patients.

Please give more information on Quantitative RT-PCR. Reporter gene used, justification of this choice. Was 2-deltadelta CT used?

In  discussion, the authors mention that In CKD, excessive renal mTORC1 activation induced by hypoxia, oxidative stress. CAn they discuss the role of anemia linked to uremic toxins in CKD to trigger this? See for example PMID: 32899941 for discussion

Author Response

Response to Reviewers' Comments

We examined the comments made by the referees carefully and prepared a revised version of our paper, in which the comments made by the referees have been addressed. I hope that the present, improved version of our manuscript will now be found to be acceptable for publication in Toxins.

Reviewer 2

Comment-1

please give specific number for animal experiment authorization. What was the sex of animals used? please justify the choice;

Reply-1

As the reviewer suggested, the specific number (A2021-021) for animal experiment authorization (page 10, line 437) and the sex of animals (page 10, line 437) were added in the Methods section of the revised manuscript.

Comment-2

Please give more information on collection and use of serum samples from CKD and normal patients.

Reply-2

As the reviewer suggested, the more information on collection and use of serum samples from CKD and normal patients was added in the Methods section of the revised manuscript as follows;

“Sixteen CKD patients between the ages of 69 and 86 years (mean 81.0 ± 1.6 years) who were undergoing dialysis and who had been admitted to the Department of Nephrology at the Akebono Clinic in Japan in October 2018 were enrolled in this study. As controls, healthy volunteers (4 males and 2 females) aged 21-29 years (mean 23.6 ± 0.7 years) were enrolled. The experimental method was described previously [42]. This study was conducted in accordance with the Declaration of Helsinki and was approved by the Ethics Committee of the Faculty of Life Sciences, Kumamoto University (approval number 1578).” (page 9, line 389-line 396)

Comment-3

Please give more information on Quantitative RT-PCR. Reporter gene used, justification of this choice. Was 2-deltadelta CT used?

Reply-3

As the reviewer suggested, more information on Quantitative RT-PCR were added in the Methods section of the revised manuscript as follows;

“The mRNA expression levels were calculated using the 2-ΔΔCt method.” (page 9, line 409-line 410)

Comment-4

In discussion, the authors mention that in CKD, excessive renal mTORC1 activation induced by hypoxia, oxidative stress. Can they discuss the role of anemia linked to uremic toxins in CKD to trigger this? See for example PMID: 32899941 for discussion.

Reply-4

As the reviewer suggested, IS also contributes to the anemia observed in CKD (Hamza E et al. Cells. 2020).

              Therefore, the following sentences and reference in addition to the additional data (Table S2) were added in the Discussion section of the revised manuscript. (page 8, line 291-line 295)

“In addition, IS also contributes to the anemia observed in CKD (Hamza E et al. Cells. 2020). In our study, the administration of AST-120 recovered the decrease of red blood cell count and hemoglobin level as observed in CKD mice (Table S2). These findings suggest that anemia may associated with IS-induced mTORC1 activation in CKD.”

Reference

  1. Hamza E, Metzinger L, Metzinger-Le Meuth V. Uremic Toxins Affect Erythropoiesis during the Course of Chronic Kidney Disease: A Review. Cells 2020;9(9)